# ORPER: A Workflow for Constrained SSU rRNA Phylogenies

**DOI:** 10.3390/genes12111741

**Published:** 2021-10-29

**Authors:** Luc Cornet, Anne-Catherine Ahn, Annick Wilmotte, Denis Baurain

**Affiliations:** 1BCCM/IHEM, Mycology and Aerobiology, Sciensano, 1050 Bruxelles, Belgium; 2BCCM/ULC Collection, InBioS–Centre for Protein Engineering, University of Liège, 4000 Liège, Belgium; acahn@uliege.be (A.-C.A.); awilmotte@uliege.be (A.W.); 3InBioS–PhytoSYSTEMS, Unit of Eukaryotic Phylogenomics, University of Liège, 4000 Liège, Belgium

**Keywords:** cyanobacteria, SSU (16S) rRNA, phylogenomics, sequencing, workflow, ribosomal proteins

## Abstract

The continuous increase in sequenced genomes in public repositories makes the choice of interesting bacterial strains for future sequencing projects ever more complicated, as it is difficult to estimate the redundancy between these strains and the already available genomes. Therefore, we developed the Nextflow workflow “ORPER”, for “ORganism PlacER”, containerized in Singularity, which allows the determination the phylogenetic position of a collection of organisms in the genomic landscape. ORPER constrains the phylogenetic placement of SSU (16S) rRNA sequences in a multilocus reference tree based on ribosomal protein genes extracted from public genomes. We demonstrate the utility of ORPER on the Cyanobacteria phylum, by placing 152 strains of the BCCM/ULC collection.

## 1. Introduction

Cyanobacteria form a phylum of bacteria, which have colonized very diversified ecosystems [1]. They are the only bacteria able to perform oxygenic photosynthesis and appeared at least 2.4 billion years ago [2]. By increasing the free atmospheric oxygen, Cyanobacteria had a critical impact on shaping life on Earth [3,4]. Beyond their ecological importance, this phylum also has an evolutionary interest due to their key role in the emergence of Archaeplastida through the primary endosymbiosis, which gave rise to the plastid [5]. Although the exact mechanisms, which include the generally accepted unicity of the event, are yet to be fully understood, it is well known that Cyanobacteria played a major role in the spread of oxygenic photosynthesis [6]. More recently, the group attracted an additional interest after uncovering, through metagenomic studies, the existence of non-photosynthetic “cyanobacteria”, notably the phylum Melainabacteria [7].

Due to this importance, the published cyanobacterial phylogenies are numerous (see for instance: [8,9,10,11,12,13,14]). The number of available genomes logically followed this interest, rising from a few hundred in 2013, when Shih et al. [15] improved the coverage of the phylum, to more than 3000 nowadays, according to GenBank statistics. Nevertheless, recent studies have demonstrated that cyanobacterial diversity, both for photosynthetic [16] and non-photosynthetic [14] representatives (when considering Melainabacteria as part of Cyanobacteria), is not well covered by the sequencing effort.

The gold standard for the estimation of bacterial diversity remains the SSU rRNA gene of the small subunit of the ribosomal RNA [17]. This locus is frequently used by scientists and culture collections to evaluate the genomic potential of newly isolated organisms. However, due to the constant and rapid growth of genome repositories, it is difficult for researchers to estimate the redundancy between these public data sources and their own collections of organisms. Here, we release ORPER, which stands for “ORganism PlacER”, an automated workflow intended to determine the phylogenetic position of organisms, for which only the SSU rRNA has been determined, in the public genomic landscape.

## 2. Methods

### 2.1. Functional Overview

The principle of ORPER is to provide an overview of the sequenced coverage (i.e., the diversity of available genomes) of a given taxon and to place SSU rRNA sequences in this diversity. ORPER first downloads the complete genomes of the taxon of interest, then extracts their ribosomal proteins to compute a reference phylogenetic tree, and finally uses this tree to constrain the backbone of a SSU rRNA phylogeny including the additional strains. The workflow uses two groups: (i) the main group corresponding to the taxonomic group of the SSU rRNA sequences (the taxon of interest) and (ii) the outgroup to the root of the phylogenetic tree. The main group is used to compute a phylogenetic tree to guide the placement of SSU rRNA sequences; therefore, it is named “reference group” for the remainder of this manuscript (Figure 1). All steps are embedded in a Nextflow script [18], and a Singularity definition file is provided for containerization [19]. ORPER is available at https://github.com/Lcornet/ORPER, accessed on 1 October 2021.

### 2.2. Workflow Details

#### 2.2.1. Taxonomy and Metadata Download

ORPER begins by creating a local copy of the National Center for Biotechnology Information (NCBI) Taxonomy [20] with the script *setup-taxdir.pl* v0.211470 from Bio::MUST::Core (D. Baurain; https://metacpan.org/dist/Bio-MUST-Core, accessed on 1 October 2021. Genome accession numbers (i.e., GCF numbers) are fetched from the NCBI Reference Sequence project (RefSeq) [21] and the taxonomy of the corresponding organisms is determined with the script *fetch-tax.pl* v0.211470 (Bio::MUST::Core package). If required, GenBank genomes [22] can be used in the same way for the both reference group and outgroup creation, independently. Four taxonomic levels are available in ORPER (phylum, class, order, family) and the user must specify the reference group and the outgroup separately (Figure 1).

#### 2.2.2. Genome Filtration and Dereplication

*CheckM* v1.1.3 with the “lineage_wf” option, is used to estimate completeness and contamination of the assemblies [23]. *Barrnap* v0.9, with default options, is used to predict rRNA genes in downloaded genomes (available at https://github.com/tseemann/barrnap, accessed on 1 October 2021). Genomes with a completeness level above 90%, a contamination level below 5%, and at least one predicted SSU rRNA sequence are retained. A dereplication step of the genomes from the reference group can be optionally carried out using *dRep* [24] and default parameters. *Prodigal*, with default options, is used to obtain conceptual proteomes [25]. All genomes from the reference group that remain after the filtration steps are used, whereas only the ten first genomes of the outgroup are used for de novo protein prediction (Figure 1).

#### 2.2.3. Reference Phylogeny Inference

Prokaryotic ribosomal protein alignments from the RiboDB database [26] are downloaded by ORPER once at the first usage. An orthologous enrichment of these alignments with sequences from the remaining proteomes (post-filtration and dereplication) is performed by *Forty-Two* v0.210570 [27,28]. These sequences are then aligned using *MUSCLE* v3.8.31 [29] in order to generate new alignment files with only the sequences from the reference group and the outgroup. Conserved sites are selected using *BMGE* v1.12 [30] with moderately severe settings (*entropy cut-off* = 0.5, *gap cut-off* = 0.2). A supermatrix is then generated using *SCaFoS* v1.30k [31] with default settings. Finally, a reference phylogenomic analysis is inferred using *RAxML* v8.2.12 [32] with 100 bootstrap replicates under the PROTGAMMALGF model.

#### 2.2.4. Constrained SSU rRNA Phylogeny

The SSU rRNA sequences provided by the user can be optionally dereplicated using *CD-HIT-EST* v4.8.1 with default parameters [33]. The SSU rRNA phylogenetic tree is inferred from both the sequences provided by the user and those extracted from the complete genomes using *RAxML* v8.2.12 [32] with 100 bootstrap replicates under the GTRGAMMA model and the phylogenomic tree as a constraint.

### 2.3. Design Considerations

ORPER compensates for the lack of phylogenetic resolution of SSU rRNA gene sequences by using ribosomal protein genes from publicly available genomes to infer a reference multilocus tree, which is then used to constrain the SSU rRNA phylogeny. Indeed, it is well known that SSU rRNA suffers, as do all single-gene phylogenies, from a lack of phylogenetic resolution [34,35,36,37,38]. Ribosomal protein genes are frequently used to perform phylogenetic placement; for instance, CheckM uses this approach to place genomes before performing the contamination estimation [23].

The NCBI databases, regularly synchronized with the European Nucleotide Archive (ENA) [39], are the most complete public databases. By default, ORPER uses only RefSeq because the latter contains only high-quality genomes [21]. Nevertheless, it might be necessary to use more genomes to estimate the actual sequence coverage of a taxon. This is especially true with metagenomic data that are, by design, not included in RefSeq [21]. That is why GenBank can be enabled as an option in ORPER. In any case, starting from a NCBI database entails the use of thousands of genomes, which can dramatically increase the computing time. For this reason, we implemented the optional use of dRep to dereplicate the genomes [24]. This allowed the user to decrease the number of genomes while conserving the sequenced diversity [24]. However, this option should be used carefully because the need for dereplication (or not) is dependent on the biological question [40]. For example, the genomic comparison of closely related strains requires using as many genomes as possible to identify individual differences. Finally, genomes in public repositories are not devoid of contamination (i.e., the inclusion of foreign DNA in the genomic data) [41,42]. Therefore, we used CheckM [23], the most commonly used tool for genomic contamination detection, and thresholds from the Genomic Standards Consortium [43] (completeness above 90% and contamination below 5%) to filter our genomes, which is a mandatory step in the workflow.

Nextflow is the latest workflow system. It was developed to increase reproducibility in science [18]. Nextflow further presents the advantage of exploiting Singularity containers as an operating system [18], which ensures the sustainability of future analyses. Singularity containers [19] correct the security issues of older container systems, thereby offering the possibility of deploying them on HPC systems where security is often an important concern. Owing to these advantages, we chose the combination Nextflow-Singularity for ORPER. Albeit ORPER is a workflow, we designed it as a program with a single command-line interface. The installation of ORPER only requires two shell commands (see https://github.com/Lcornet/ORPER, accessed on 1 October 2021). Moreover, the analysis reported in this study can be replayed with a single command in less than one day using 30 CPU cores (Intel Xeon E5-2640 v4 series) (see https://github.com/Lcornet/ORPER, accessed on 1 October 2021).

## 3. Results and Discussion

### Case Study: BCCM/ULC Cyanobacteria Collection

Phylogenomic studies of Cyanobacteria are numerous, notably focusing on the emergence of multicellularity [44,45,46], the appearance of oxygenic photosynthesis [47,48,49], or the origin of plastids [9,50,51,52]. The emergence of the plastid remains quite unclear [6] with potential origins either among heterocyst-forming cyanobacteria [2,53] or earlier diverging lineages [9,50,51,52]. Therefore, the selection of Cyanobacteria for future sequencing projects remains an important issue.

We tested ORPER on this phylum with 152 SSU rRNA sequences from the BCCM/ULC collection. The information on the SSU rRNA sequences and the collection itself is available in Appendix A. RefSeq genomes for the “Cyanobacteria” phylum were specified as the reference group, whereas genomes for the “Melainabacteria” phylum available in GenBank were used as the outgroup (Figure 2). The dereplications for the reference genomes and for the SSU rRNA sequences were both activated. The reference tree inferred by ORPER was based on a supermatrix of 372 organisms × 6246 unambiguously aligned amino-acid positions (7.92% missing character states). The 152 SSU rRNA input sequences used in this study were dereplicated to 140 sequences at a 95% identity threshold, and were then used to compute the constrained tree.

We chose to compare the phylogeny inferred by ORPER to the latest multilocus (ribosomal) phylogeny of the cyanobacterial phylum published by Moore et al. (2019), who identified the earliest potential basal position of the plastids [9]. The constrained tree computed by ORPER is comparable to the tree of Moore et al. (2019), with ten out of eleven clades recovered by ORPER (Figure 2). The only missing clade, clade 7, was represented by genomes neither present in RefSeq [9] nor in the ULC strains, and thus was logically absent from our phylogeny. The 140 BCCM/ULC strains obtained after dereplication covered the whole diversity of publicly available cyanobacterial genomes. Three BCCM/ULC strains (ULC415, ULC417, ULC381) formed a basal clade clustered with *Limnothrix* sp. GCF_002742025.1, which was not present in Moore et al.’s analysis (Appendix A). These three strains are, therefore, of high interest for genome sequencing, especially in the context of plastid emergence. Here, we analyzed the cyanobacterial phylum, but ORPER could be used on any bacterial taxon of the NCBI.

## 4. Conclusions

ORPER is a state-of-the-art tool, designed for the phylogenetic placement of SSU rRNA sequences in a phylogenetic tree constrained by a multilocus tree. We demonstrated the utility of ORPER on Cyanobacteria, using sequences from the BCCM/ULC collection, to estimate the phylogenetic position of SSU rRNA sequences among the landscape of sequenced genomes. Its easy-to-use installation process and Singularity containerization makes ORPER a useful tool for culture collections and for scientists to use in their future selection of genomes to sequence.

## Figures and Tables

**Figure 1 genes-12-01741-f001:**
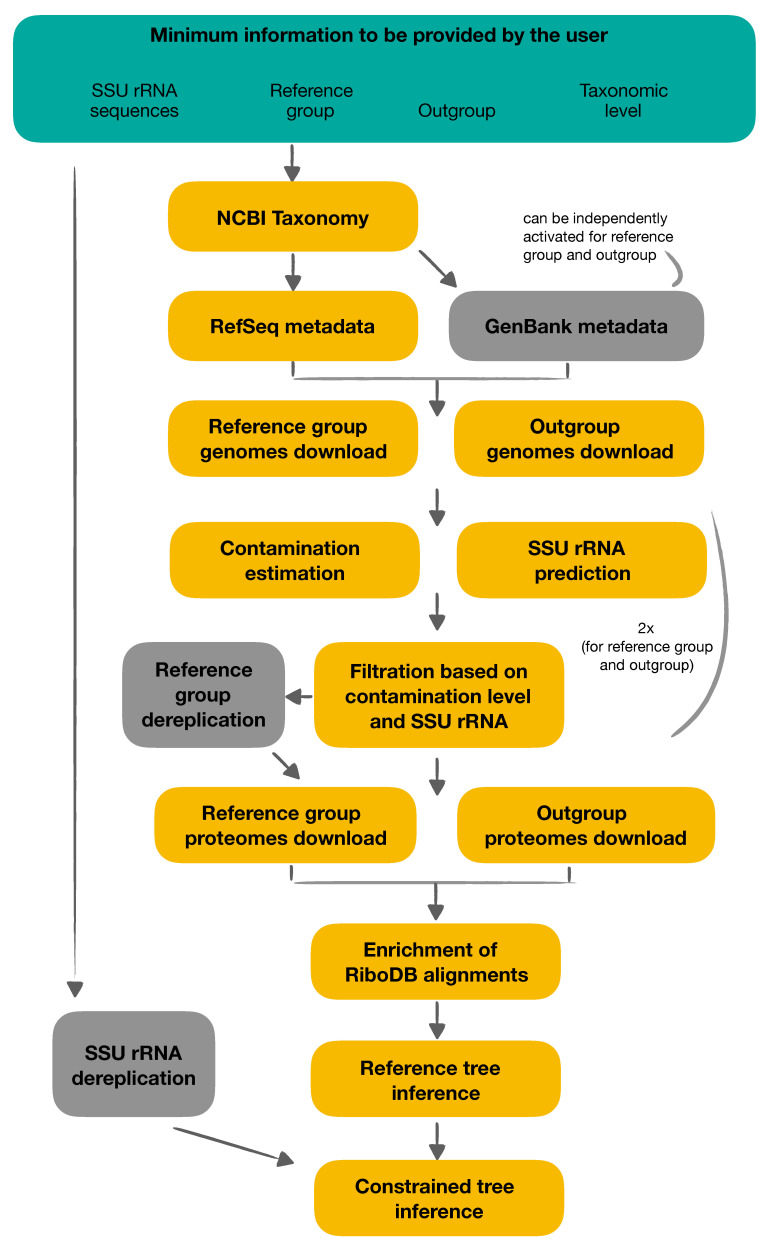
Overview of ORPER workflow. Users should specify at least four pieces of information to run ORPER: (i) their SSU (16S) rRNA sequences, (ii) the taxon of interest, (iii) the outgroup of the phylogeny and (iv) the taxonomic level (Green part). Yellow boxes are mandatory steps of ORPER whereas grey boxes are optional steps. Contamination estimation, SSU rRNA prediction and filtration are performed twice, once for the reference group and once for the outgroup.

**Figure 2 genes-12-01741-f002:**
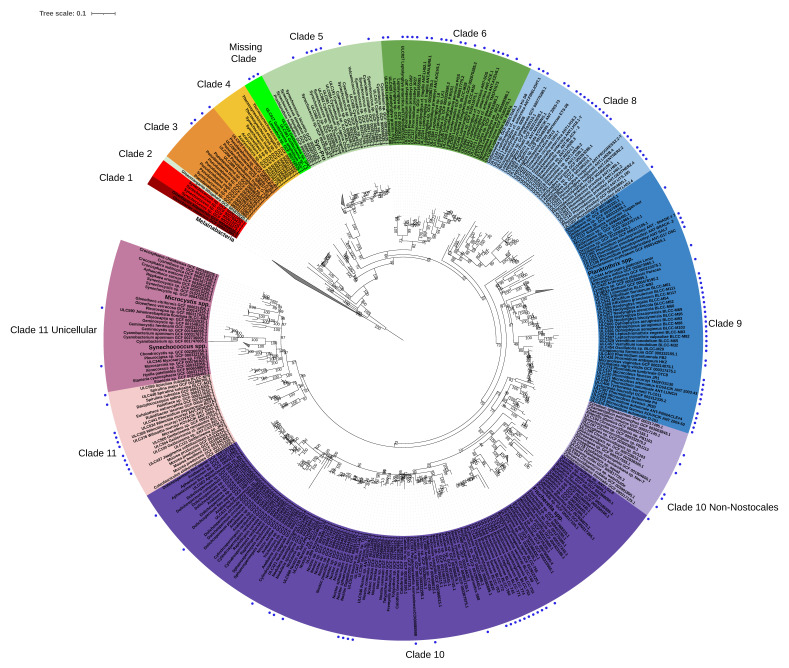
Constrained cyanobacterial phylogenetic tree of the BCCM/ULC collection. The tree is the output of ORPER, a Maximum-likelihood constrained inference computed under the GTRGAMMA model. Clades correspond to the groups defined in Moore et al. (2019) [9]. Clades 10 and 11 have been divided into two sub-clades, adding, respectively “Non-Nostocales” and “Unicellular” sub-clades to Moore et al.’s phylogeny. Blue dots indicate ULC/BCCM strains. The clade absent from Moore et al.’s phylogeny is indicated as “Missing Clade”.

## Data Availability

ORPER is freely available at https://github.com/Lcornet/ORPER (accessed on 1 October 2021).

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
