# Peer review of "ORPER: A Workflow for Constrained SSU rRNA Phylogenies"

_genes, 2021, doi:10.3390/genes12111741_

Round 1

Reviewer 1 Report

The manuscript “ORPER: a workflow for constrained SSU rRNA phylogenies” reports the development of a program that aims to help genome sequencing projects to decide the most interesting bacterial targets according to taxonomic coverage in public databases. The workflow is based on placing the 16S rRNA gene sequences of target organisms in a multilocus tree reconstructed from ribosomal proteins encoded in publicly available genomes. This was demonstrated with a case study on sequences from the bacterial phylum Cyanobacteria.

This program is quite interesting and reduces the amount of time necessary for deciding which organisms should be prioritize in genomic research. In my opinion, the article is relevant and adequately describes the software and its workflow. However, I recommend a minor revision to improve a few points as follows.

Line 27: “Eukaryotic algae” is a redundant term as there are no prokaryotic algae. Additionally, Archaeplastida includes more than just algae. I suggest to focus on the more inclusive and coherent group, Archaeplastida.

Line 33: I don’t think it is appropriate to cite Soo et al. (2017) in relation to the discovery of Melainabacteria, especially in light of the fact that these authors have recently renamed this candidate taxon to Vampirovibrionia (10.1016/j.freeradbiomed.2019.03.029). The most appropriate citation here would be the original Di Rienzi et al. (2013) article (10.7554/eLife.01102). In addition, this sentence makes it sound like all non-photosynthetic bacteria closely related to the cyanobacteria were classified under melainabacteria, which is inaccurate since the discovery of Sericytochromatia, Margulisbacteria and Saganbacteria.

Lines 41-42 and elsewhere: “SSU (16S)” was unnecessarily repeated throughout the manuscript. There is no need keep using both terms. I suggest to keep it either as 16S rRNA or, like in Figure 1, SSU rRNA.

Lines 102 and 108: Why were the PROTGAMMALGF and GTRGAMMA chosen? Why were predetermined models preferred instead of adding a model selection step to the pipeline?

Line 150: I wouldn’t say the origin of plastids is a controversial topic based merely on an article that explores hypothetical, alternative scenarios. The details are indeed still unclear (e.g. what cyanobacterial branch originated them and the depth of horizontal gene transfer that occurred), but broadly speaking there is a large degree of consensus in the field. That is an overstatement.

Line 154: Since this paper is intended for a broader audience and not just cyanobacteriologists, I would provide the address for the BCCM/ULC website and some other information for those unfamiliar with this collection.

Lines 154-155: How were the 16S rRNA gene sequences obtained?

Line 156: In line 33 Melainabacteria was mentioned as a class, but here they were considered a phylum. Of course one can choose either side of this ongoing discussion, but it is important to keep it consistent. Since Melainabacteria are being cited instead of Vampirovibrionia I would keep phylum throughout.

In Figure 1, it is unclear to me which steps are carried out 2x for reference group and outgroup. Please clarify which procedures are being repeated.

Figure 2 is very hard to read and the characters are misaligned. Perhaps the proofing system is messing with the formatting of the vector file. Please double check that this figure, maybe supplying it under another format (like pdf) or even turning it into a raster image like Figure 1 instead. Also, the clade that was not present in Moore et al. (2019) could also be highlighted for the reader since that is the most important part of the tree according to the last paragraph in the results.

In Supplemental File 1, please add accession numbers for the 16S rRNA gene sequences of the strains used in this work.

Author Response

The manuscript “ORPER: a workflow for constrained SSU rRNA phylogenies” reports the development of a program that aims to help genome sequencing projects to decide the most interesting bacterial targets according to taxonomic coverage in public databases. The workflow is based on placing the 16S rRNA gene sequences of target organisms in a multilocus tree reconstructed from ribosomal proteins encoded in publicly available genomes. This was demonstrated with a case study on sequences from the bacterial phylum Cyanobacteria.

This program is quite interesting and reduces the amount of time necessary for deciding which organisms should be prioritize in genomic research. In my opinion, the article is relevant and adequately describes the software and its workflow. However, I recommend a minor revision to improve a few points as follows.

  • We thank the reviewer for detailed reading of our article and judicious comments. We did our best to integrate the recommendations, please see below our answers.

Line 27: “Eukaryotic algae” is a redundant term as there are no prokaryotic algae. Additionally, Archaeplastida includes more than just algae. I suggest to focus on the more inclusive and coherent group, Archaeplastida.

  • This is a matter of taste, but distracting here. Therefore, the sentence has been changed to focus only on Archaeplastida.

Line 33: I don’t think it is appropriate to cite Soo et al. (2017) in relation to the discovery of Melainabacteria, especially in light of the fact that these authors have recently renamed this candidate taxon to Vampirovibrionia (10.1016/j.freeradbiomed.2019.03.029). The most appropriate citation here would be the original Di Rienzi et al. (2013) article (10.7554/eLife.01102). In addition, this sentence makes it sound like all non-photosynthetic bacteria closely related to the cyanobacteria were classified under melainabacteria, which is inaccurate since the discovery of Sericytochromatia, Margulisbacteria and Saganbacteria.

  • The sentence has been changed to mention Melaninabacteria as part of non-photosynthetic cyanobacteria. The citation of Soo et al., 2017 has been changed for Di Rienzi et al. (2013).

Lines 41-42 and elsewhere: “SSU (16S)” was unnecessarily repeated throughout the manuscript. There is no need keep using both terms. I suggest to keep it either as 16S rRNA or, like in Figure 1, SSU rRNA.

  • The term 16S has been deleted in the manuscript after first occurrence in the main text.

Lines 102 and 108: Why were the PROTGAMMALGF and GTRGAMMA chosen? Why were predetermined models preferred instead of adding a model selection step to the pipeline?

  • RAxML offers few choices of models, hence the choice of these two models is a pragmatic one. This was sufficient for the purpose of ORPER who is to compute simple phylogenetic analyses to facilitate the placement of SSU rRNA sequences. Nevertheless, we will make ORPER evolve in the future according to the demands of users, for example by including multilocus-based reconstruction with IQ-TREE and its more numerous models

Line 150: I wouldn’t say the origin of plastids is a controversial topic based merely on an article that explores hypothetical, alternative scenarios. The details are indeed still unclear (e.g. what cyanobacterial branch originated them and the depth of horizontal gene transfer that occurred), but broadly speaking there is a large degree of consensus in the field. That is an overstatement.

  • Fair enough. The sentence has been changed to correct the overstatement.

Line 154: Since this paper is intended for a broader audience and not just cyanobacteriologists, I would provide the address for the BCCM/ULC website and some other information for those unfamiliar with this collection.

  • The website and explanations on BCCM/ULC are added to the Supplementary file 1.

Lines 154-155: How were the 16S rRNA gene sequences obtained?

  • A paragraph has been added to Supplementary file 1, describing how the 16S rRNA sequences were obtained.

Line 156: In line 33 Melainabacteria was mentioned as a class, but here they were considered a phylum. Of course one can choose either side of this ongoing discussion, but it is important to keep it consistent. Since Melainabacteria are being cited instead of Vampirovibrionia I would keep phylum throughout.

  • The sentence on line 33 has been changed to mention Melainabacteria as a phylum.

In Figure 1, it is unclear to me which steps are carried out 2x for reference group and outgroup. Please clarify which procedures are being repeated.

  • Legends of Figure 1 has been lengthened: “Contamination estimation, SSU rRNA prediction, and filtration are performed twice, once for the reference group and once for the outgroup.”

Figure 2 is very hard to read and the characters are misaligned. Perhaps the proofing system is messing with the formatting of the vector file. Please double check that this figure, maybe supplying it under another format (like pdf) or even turning it into a raster image like Figure 1 instead. Also, the clade that was not present in Moore et al. (2019) could also be highlighted for the reader since that is the most important part of the tree according to the last paragraph in the results.

  • The clade not present in Moore et al. (2019) is now indicated in Figure 2. According to reviewer 1 and 2 comments, we decided to add a vertical representation of this figure in Supplementary File 2 to help the visualization.

In Supplemental File 1, please add accession numbers for the 16S rRNA gene sequences of the strains used in this work.

  • The accessions of SSU rRNA sequences were added in Supplemental File 1.

Reviewer 2 Report

Review of the MS entitled “ORPER: A Workflow for Constrained SSU rRNA Phylogenies” by Cornet L et al. submitted to Genes.

The authors did develop a workflow that allows using a series of programs to obtain a constrained 16S rRNA phylogeny of their organisms of interest. The input of the user consists in their studied 16S rRNA sequences, the choice of the in- and outgroup and the taxonomic level of the study that goes from phyla to family. The information for the constraints of the16S rRNA phylogeny is obtained through the analysis of the concatenated ribosomal proteins. Furthermore, in the case of a densely sampled taxon, the user has the possibility to reduce the number of studied taxa via dereplication.

The paper is well written and concise, but contains nevertheless a number of highly interesting references. The subject is timely to help researchers to cope with a true avalanche of newly established 16S rRNA sequences and to choose the best organisms for genome sequencing.

There are no major points.

I have a few suggestions and some typos.

It would be better to give the explanation of the term ORPER (Organism Placer) at the beginning a good place would be the abstract.

The circular form of Figure 2 has without any doubt certain merits. One of the problems that I have with this kind of figures is that it is quite tedious to get a detailed overview of this tree. My proposition is therefore to add a vertical version of it to the supplements.

There are two typos in Reference 16 and 38 in form of “RRNA”.

Reference 40 is incomplete.

On line 154 it would be better to say in this case “we tested … with”.

On line 168 it would be better to use “neither … nor”.

Author Response

Review of the MS entitled “ORPER: A Workflow for Constrained SSU rRNA Phylogenies” by Cornet L et al. submitted to Genes.

The authors did develop a workflow that allows using a series of programs to obtain a constrained 16S rRNA phylogeny of their organisms of interest. The input of the user consists in their studied 16S rRNA sequences, the choice of the in- and outgroup and the taxonomic level of the study that goes from phyla to family. The information for the constraints of the16S rRNA phylogeny is obtained through the analysis of the concatenated ribosomal proteins. Furthermore, in the case of a densely sampled taxon, the user has the possibility to reduce the number of studied taxa via dereplication.

The paper is well written and concise, but contains nevertheless a number of highly interesting references. The subject is timely to help researchers to cope with a true avalanche of newly established 16S rRNA sequences and to choose the best organisms for genome sequencing.

  • We thank the reviewer for this positively oriented review. We did our best to address the comments, please see below our answers.

There are no major points.

I have a few suggestions and some typos.

It would be better to give the explanation of the term ORPER (Organism Placer) at the beginning a good place would be the abstract.

  • The explanation has now been moved into the abstract.

The circular form of Figure 2 has without any doubt certain merits. One of the problems that I have with this kind of figures is that it is quite tedious to get a detailed overview of this tree. My proposition is therefore to add a vertical version of it to the supplements.

  • A vertical representation of Figure 2 has been added in supplementary file 2.

There are two typos in Reference 16 and 38 in form of “RRNA”.

  • The references are now correct.

Reference 40 is incomplete.

  • The reference is now correct.

On line 154 it would be better to say in this case “we tested … with”.

  • The sentence has been changed.

On line 168 it would be better to use “neither … nor”.

  • The sentence has been changed.